# InstaTrain: Adaptive Training via Ultra-Fast Natural Annealing within Dynamical Systems

**Chuan Liu, Ruibing Song, Chunshu Wu, Pouya Haghi, Tony (Tong) Geng**
University of Rochester, Rochester, NY, USA
`{chuan.liu, ruibing.song, chunshu.wu, tong.geng}@rochester.edu`
`{phaghi}@ur.rochester.edu`

## Abstract

Time-series modeling is broadly adopted to capture underlying patterns present in historical data, allowing prediction of future values. However, one crucial aspect of such modeling is often overlooked: in highly dynamic environments, data distributions can shift drastically within a second or less. Under these circumstances, traditional predictive models, and even online learning methods, struggle to adapt to the ultra-fast and complex distribution shifts present in highly dynamic scenarios. To address this, we propose **InstaTrain**, a novel learning approach that enables ultra-fast model updates for real-world prediction tasks, thereby keeping pace with rapidly evolving data distributions. In this work, (1) we transform the slow and expensive training process into an ultra-fast natural annealing process within a dynamical system. (2) Leveraging a recently proposed electronic dynamical system, we augment the system with parameter update modules, extending its capabilities to encompass both rapid training and inference. Experimental results on highly dynamic datasets demonstrate that our method achieves orders-of-magnitude improvements in training speed and energy efficiency while delivering superior accuracy compared to baselines running on GPUs.

## 1 Introduction

Time-series prediction lies at the heart of artificial intelligence, powering applications ranging from weather forecasting (Karevan & Suykens, 2020; Bochenek & Ustrnul, 2022) to product and content recommendation (He et al., 2017; Zhang et al., 2021). Current neural network methods have achieved remarkable success by learning the joint distribution between inputs and predictions (Lim & Zohren, 2021). However, these methods often implicitly assume that the learned joint distribution remains stable over a considerably long period, an assumption that can easily be violated when the underlying distribution undergoes severe shifts, consequently causing significant failures in trained models. In response to this challenge, the community has pivoted toward more adaptive learning strategies, such as online learning and continual learning approaches (Hoi et al., 2021; Chen et al., 2021; Pham et al., 2022; Wen et al., 2024). These methodologies are designed to incrementally adjust model parameters, thereby maintaining alignment with current data trends. Despite their advancements, they struggle to adapt to the circumstances where data distribution evolves rapidly due to their insufficient adaptation speed. This underscores the pressing need for more agile and responsive approaches that can swiftly adapt to shifts in distribution and ensure model effectiveness.

In the post-Moore's Law era, the limitations of speed improvements in digital processors (such as CPUs and GPUs) have become more pronounced, attracting growing attention in novel computing substrates, a promising yet largely untapped area of research (Momeni et al., 2024). As a promising candidate, a recently developed electronic dynamical system (Afoakwa et al., 2021; Sharma et al., 2022) stands out due to its timely applicability, low power consumption, and exceptionally fast computational speed. Similar to natural dynamical systems, where particles naturally move toward lower energy states following the Second Law of Thermodynamics, the behavior of this electronic dynamical system is governed by its energy function (Hamiltonian), with lower energy states being rapidly reached through **natural annealing** – a process where electrons spontaneously move among capacitors to seek equilibrium at the "speed of electrons" and on the milliwatt scale.

However, despite this system having been utilized to accelerate graph learning inference (Wu et al., 2024), the model training process still relies on conventional digital processors, which cannot keep up with rapidly evolving data distributions in real-world applications. Consequently, a more advanced learning approach is critically needed to fully exploit the exceptional computational power of this dynamical system. Since the system specializes in performing natural annealing, it holds the potential to meet the ultra-fast model learning demand if the sluggish offline training process can be transformed into the natural annealing process. This idea also aligns with the concept of "mortal computation" proposed by (Hinton, 2022), which advocates for the integration of algorithms and hardware, thereby offering significantly lower costs compared to conventional neural networks running on CPUs and GPUs.

In response to this opportunity, we propose **InstaTrain**, which extends the extraordinary computational efficiency of the electronic dynamical system from inference to training, addressing the need for capturing rapidly evolving data distributions. The overall framework of this approach is illustrated in Fig. 1, comprising two major components. **(1) Training Algorithm:** Formulated as a dynamical system, our model is determined by the trainable parameters in the Hamiltonian, or energy function. The proposed algorithm accomplishes training through an iterative natural annealing process, which pushes the lowest energy state of the dynamical system to match the ground truth provided by training data. **(2) Hardware Augmentation:** We enhance the dynamical system

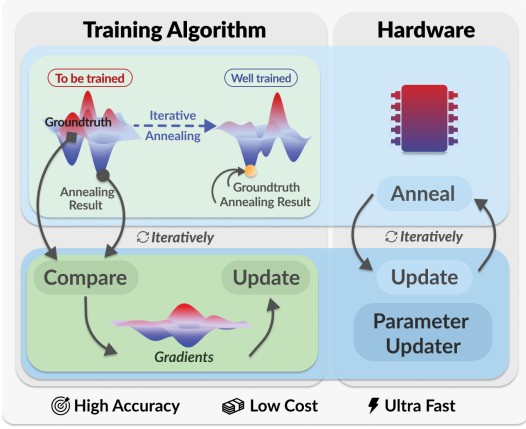

Figure 1: Overview of InstaTrain.

with parameter update modules to realize the self-training mechanism. This allows both training and inference to be carried out on the same hardware, resulting in outstanding computational efficiency and, essentially, achieving real-time model adaptation upon highly dynamic data distributions.

The core contributions of this paper are summarized as follows:

- We propose InstaTrain, a novel learning approach designed to address the demands for ultra-fast model adaptation in highly dynamic time-series prediction.

- We transform the sluggish offline training into an iterative natural annealing process within a dynamical system, enabling ultra-fast model training and updating.

- We augment the original dynamical system with parameter update modules, extending its capabilities to encompass both rapid training and inference.

- Experimental results across highly dynamic datasets show that the proposed method achieves orders-of-magnitude improvements in training speed and energy efficiency while delivering superior accuracy compared to baselines running on GPUs.

## 2 BACKGROUND

This section provides background on the targeted electronic dynamical system, discussing its mathematical model, physical embodiment, and offline training method.

**Mathematical Model.** Mathematically, the dynamical system describes how components (nodes) interact and influence each other's states over time, driving the system's evolution toward equilibrium. The system has been utilized to accelerate graph learning inference (Wu et al., 2024), with a real-valued Hamiltonian function:

$$\mathcal{H}(\mathbf{s}) = -\sum_{i \neq j}^{N} J_{ij}\sigma_i\sigma_j + \sum_{i}^{N} h_i\sigma_i^2, \tag{1}$$

where $\sigma_i \in \mathbb{R}$. $\mathbf{s} = \{\sigma_1, \sigma_2, ..., \sigma_N\}$ denotes the nodes in the dynamical system, $J_{ij}$ represents the relationship between node $\sigma_i$ and node $\sigma_j$, and $h_i$ refers to the self-reaction strength, and is forced

positive. This Hamiltonian function $\mathcal{H}$ is derived from the classic binary Ising Hamiltonian (Cipra, 1987) rooted in ferromagnetism physics, but extends it to overcome the binary limitations of nodes restricted to values of $+1$ or $-1$. Specifically, the binary limitation of the Ising model refers to the failure of naively extending its binary nodes to real values. The Hamiltonian of binary Ising model is $\mathcal{H}_b(\mathbf{s}) = -\sum_{i \neq j}^{N} J_{ij}\sigma_i\sigma_j - \sum_i^{N} h_i\sigma_i$. If nodes are real-valued, they evolve to $\pm\infty$ to pursue the lowest energy state, which is $-\infty$. Even if boundaries are applied to $\sigma_i$, $\sigma_i$ is only intercepted along its way to infinity, resulting in polarized values and essentially a binary model. In contrast, in the real-valued Hamiltonian function (i.e., $\mathcal{H}$), the quadratic term acts as an energy regulator, which prevents the energy from going down to $-\infty$, allowing nodes to be localized at certain values. This extension gives nodes the ability to take on real values, thus making it feasible to perform more precise modeling of real-valued systems in real life. In this work, the Hamiltonian that supports real-valued nodes $\mathcal{H}$ is employed.

**Physical Embodiment.** The dynamical system is physically realized as an electronic circuit (Afoakwa et al., 2021), where electronic components facilitate the system's spontaneous energy decrease toward equilibrium (i.e., $d\mathcal{H}/dt \leq 0$). Specifically, each node $\sigma_i$ is represented as a voltage on a capacitor $C$, while coupling parameters $J$ and $h$ are implemented as resistor conductance. According to Lyapunov stability analysis, the node dynamics can be designed as $d\sigma_i/dt \propto -\partial\mathcal{H}_{rv}/\partial\sigma_i$:

$$\frac{d\sigma_i}{dt} = \frac{1}{C}\left(\sum_{j \neq i}^{N}(J_{ij} + J_{ji})\sigma_j - 2h_i\sigma_i\right), \tag{2}$$

guaranteeing the system evolves toward lower energy state:

$$\frac{d\mathcal{H}}{dt} = \sum_i^{N}\frac{\partial\mathcal{H}}{\partial\sigma_i}\frac{d\sigma_i}{dt} = -\sum_i^{N}\frac{1}{C}\left(\frac{\partial\mathcal{H}}{\partial\sigma_i}\right)^2 \leq 0. \tag{3}$$

Here, $C$ denotes the capacitance, a positive constant. The node dynamics indicate that the value of each node $\sigma_i$ is influenced by its input electric currents $(J_{ij} + J_{ji})\sigma_j$ from others nodes as well as its local current $2h_i\sigma_i$. Therefore, through charging or discharging the capacitors at "speed of electrons", the node values adjust rapidly, driving the system toward equilibrium with minimal power consumption.

**Offline Training.** To train the parameters $J$ and $h$ in $\mathcal{H}$, prior research (Wu et al., 2024) employs a conditional likelihood method implemented on conventional digital processors. This approach focuses on one node $\sigma_i$ at a time, treating other nodes as conditions. An estimated value for node $\sigma_i$ is computed as $\hat{\sigma_i} = \frac{1}{2h_i}\sum_{j \neq i}^{N}(J_{ij} + J_{ji})\sigma_j$. The difference between $\hat{\sigma_i}$ and the corresponding ground truth is then evaluated using loss metrics such as Mean Absolute Error (MAE). By minimizing these losses, the parameters are optimized to align the ground truth with the system's lowest energy state. Consequently, during inference, the inherent process of spontaneous energy decrease drives the system toward the lowest energy state, producing the desired solution.

# 3 METHODOLOGY: INSTATRAIN

In this section, we present InstaTrain, a novel learning approach that leverages the natural annealing process within a dynamical system to enable ultra-fast model training, capturing rapidly evolving data distribution for prediction tasks. We first introduce our Iterative Natural Annealing Training (INAT) algorithm, including how to formulate the prediction problem using the dynamical system and the detailed training process. Furthermore, we augment the original electronic hardware, integrating update modules to enable the self-training feature.

## 3.1 ITERATIVE NATURAL ANNEALING TRAINING (INAT)

### 3.1.1 FORMULATING PREDICTION VIA NATURAL ANNEALING

Consider a time-series prediction task where we aim to predict a system's future states $\mathbf{s}^{t+1}$ based on its historical states $\mathbf{s}^t$, i.e., $\mathbf{s}^{t+1} = f_\theta(\mathbf{s}^t)$. The goal is to optimize the parameters $\theta$ so that $f_\theta$

accurately captures the system's evolution over time. We can reformulate this prediction problem using a dynamical system. Specifically, let $\mathbf{s}^t$ and $\mathbf{s}^{t+1}$ represent the node configurations of the dynamical system at consecutive time steps. Without loss of generality, we clamp the first $N/2$ node values to the input state $(\sigma_1, ..., \sigma_{N/2}) = \mathbf{s}^t$, and allow the remaining $N/2$ nodes $(\sigma_{N/2+1}, ..., \sigma_N)$ to anneal freely. If the Hamiltonian parameters $J$ and $h$ perfectly capture the dependencies between inputs and predictions, then the ground truth configuration $\mathbf{s}^* = (\mathbf{s}^t, \mathbf{s}^{t+1})$ corresponds to the lowest energy of the dynamical system. Consequently, by clamping the input nodes to $\mathbf{s}^t$ and letting the remaining nodes evolve according to the designed dynamics (Eq. 2), the natural annealing process will drive the system toward the lowest energy state (equilibrium), resulting in the remaining nodes converging to the desired solution $\mathbf{s}^{t+1}$.

We can further interpret this annealing process using the Boltzmann distribution, which defines a mapping from energy to probability. Specifically, the lowest energy node configuration corresponds to the maximum probability state through the following:

$$p_{\mathbf{s}^*} = \frac{1}{Z} e^{-\mathcal{H}(\mathbf{s}^*)}, \tag{4}$$

where $Z$ is the partition function defined as $\int e^{-\mathcal{H}} d\sigma$, functioning as a normalizing constant. Therefore, the system's evolution towards the lowest energy state is equivalent to finding the desired prediction $\mathbf{s}^{t+1}$ with the highest probability under the Hamiltonian $\mathcal{H}$. To elucidate more clearly, we visualize the whole process in Fig. 2. Clamping the input $\mathbf{s}^t$ confines the entire energy landscape to a subspace compatible with the given input data. Within this constrained landscape, the remaining unclamped nodes undergo natural annealing, spontaneously evolving toward the lowest energy state. This process efficiently yields the desired solution $\mathbf{s}^{t+1}$, leveraging the extraordinary computational power inherent in the dynamical system.

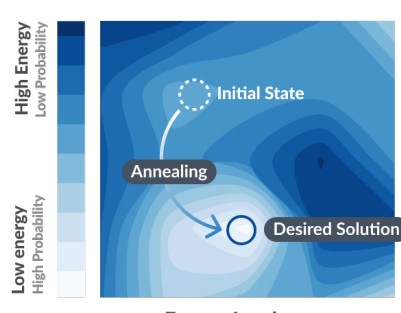

Figure 2: Prediction via annealing.

### 3.1.2 TRAINING THROUGH ITERATIVE NATURAL ANNEALING

Through the above description, we can perform efficient prediction on the dynamical system given the optimal Hamiltonian parameters $J$ and $h$. In terms of training, instead of undergoing costly training processes on digital processors, it is much more preferable that model training is also available on the dynamical system. To address this, we describe how to obtain the target parameters from the training data through an iterative natural annealing process, the same process used for inference.

Specifically, we seek to maximize the likelihood of the training set under the model:

$$\underset{J,h}{\arg\max} \prod_{\mathbf{s} \in T} p_{\mathbf{s}}, \tag{5}$$

where $T$ is the training set. This is equivalent to minimizing the negative log-likelihood loss:

$$\underset{J,h}{\arg\min} \mathcal{L}(\mathbf{s}; J, h) = \frac{1}{M} \sum_{\mathbf{s} \in T} \left( ln(Z) - ln\left(e^{-\mathcal{H}}\right) \right), \tag{6}$$

where $M$ is the number of training samples. The gradients of $\mathcal{L}$ with respect to $J_{ij}$ are given by

$$\frac{\partial \mathcal{L}(\mathbf{s})}{\partial J_{ij}} = \frac{\partial \ln(Z)}{\partial J_{ij}} + \frac{1}{M} \sum_{\mathbf{s} \in T} \frac{\partial \mathcal{H}}{\partial J_{ij}}, \tag{7}$$

where the two terms are essentially expectations of node multiplications:

$$\frac{\partial \ln(Z)}{\partial J_{ij}} = \frac{1}{Z} \frac{\partial Z}{\partial J_{ij}} = \frac{\int e^{-\mathcal{H}} \sigma_i \sigma_j \, d\sigma}{\int e^{-\mathcal{H}} \, d\sigma} = \langle \sigma_i \sigma_j \rangle_{\text{model}}, \tag{8}$$

$$\frac{1}{M} \sum_{\mathbf{s} \in T} \frac{\partial \mathcal{H}}{\partial J_{ij}} = -\frac{1}{M} \sum_{\mathbf{s} \in T} \sigma_i \sigma_j = -\langle \sigma_i \sigma_j \rangle_{\text{data}}. \tag{9}$$

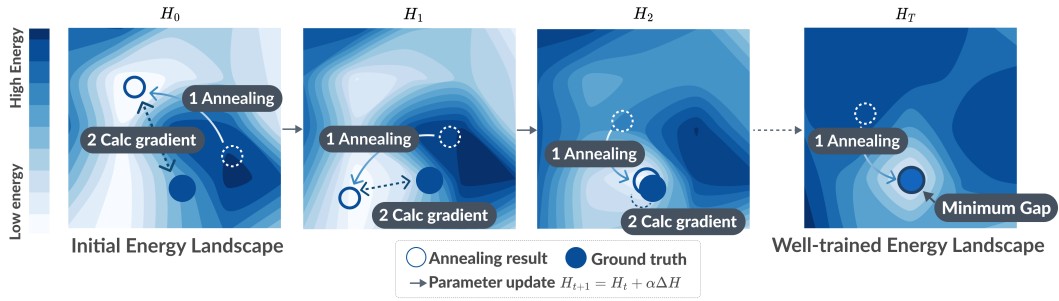

Figure 3: Model training through iterative natural annealing.

Particularly, $\langle \sigma_i \sigma_j \rangle_{\text{data}}$ denotes the expectation over the training data, which is tractable, and $\langle \sigma_i \sigma_j \rangle_{\text{model}}$ corresponds to the expectation of $\sigma_i \sigma_j$ given by the current model. Consequently, the gradient for the coupling parameter $J_{ij}$ is

$$\frac{\partial \mathcal{L}(\mathbf{s})}{\partial J_{ij}} = \langle \sigma_i \sigma_j \rangle_{\text{model}} - \langle \sigma_i \sigma_j \rangle_{\text{data}}. \tag{10}$$

In the same way, the gradients of $h_i$ are given by

$$\frac{\partial \mathcal{L}(\mathbf{s})}{\partial h_i} = \frac{\partial ln(Z)}{\partial h_i} + \frac{1}{M} \sum_{\mathbf{s} \in T} \frac{\partial \mathcal{H}}{\partial h_i} = \frac{\partial ln(Z)}{\partial h_i} + \frac{1}{M} \sum_{\mathbf{s} \in T} \sigma_i^2 = -\langle \sigma_i^2 \rangle_{\text{model}} + \langle \sigma_i^2 \rangle_{\text{data}}. \tag{11}$$

Therefore, to update the parameters, we need $\langle \sigma_i \sigma_j \rangle_{\text{model}}$ and $\langle \sigma_i^2 \rangle_{\text{model}}$, which correspond to the expectation under the current model parameters. To estimate $\langle \cdot \rangle_{\text{model}}$, we leverage the computational power of the electronic dynamical system to achieve remarkable efficiency. As described in §3.1.1, we can estimate the current model's prediction $\hat{\mathbf{s}}^{t+1}$ through clamping $\mathbf{s}^t$ to input nodes and allowing the dynamical system to perform natural annealing. By measuring the node configurations at the end of the annealing process, we can estimate the required model expectations $\langle \sigma_i \sigma_j \rangle_{\text{model}}$ and $\langle \sigma_i^2 \rangle_{\text{model}}$. In this way, the training process is transformed into an iterative natural annealing process, as described in Algorithm 1 and illustrated in Fig. 3. This innovative training process eliminates the need for computationally expensive offline training. Instead, it harnesses the natural energy decrease feature to perform efficient computations, enabling ultra-fast model training.

To summarize, the outcome of the natural annealing process depends on the accuracy of the current Hamiltonian parameters in capturing the dependencies between the inputs and predictions. When the values of some nodes are fixed to $\mathbf{s}^t$, two scenarios can occur: (1) If the parameters properly

---

**Algorithm 1** Iterative Natural Annealing Training

**Input:** Training set $T = \{\mathbf{s}_1, \mathbf{s}_2, \ldots, \mathbf{s}_M\}$, initial $J^0, h^0$, learning rate $\eta$, and training epochs $N_{\text{iter}}$.
**Output:** Trained Hamiltonian parameters $J, h$.
1: Initialize $J \leftarrow J^0, h \leftarrow h^0$.
2: **for** $i = 1$ to $N_{\text{iter}}$ **do**
3:     **for** each $\mathbf{s}_j = (\mathbf{s}_j^t, \mathbf{s}_j^{t+1})$ in $T$ **do**
4:         Clamp the first half nodes to $\mathbf{s}_j^t$
5:         Perform natural annealing to obtain $\hat{\mathbf{s}}_j^{t+1}$
6:         Get $\langle \sigma_i \sigma_j \rangle_{\text{model}}$ and $\langle \sigma_i^2 \rangle_{\text{model}}$ based on $\mathbf{s}_j^t, \hat{\mathbf{s}}_j^{t+1}$
7:         Get $\langle \sigma_i \sigma_j \rangle_{\text{data}}$ and $\langle \sigma_i^2 \rangle_{\text{data}}$ based on $\mathbf{s}_j^t, \mathbf{s}_j^{t+1}$
8:         Update $J_{ij} \leftarrow J_{ij} - \eta \cdot \langle \sigma_i \sigma_j \rangle_{\text{model}} - \langle \sigma_i \sigma_j \rangle_{\text{data}})$
9:         Update $h_i \leftarrow h_i - \eta \cdot (-\langle \sigma_i^2 \rangle_{\text{model}} + \langle \sigma_i^2 \rangle_{\text{data}})$
10:     **end for**
11: **end for**
12: **return** $J, h$

---

describe the dependencies, the annealing process will converge to the desired solution $\mathbf{s}^{t+1}$, representing the ideal case where the model has successfully learned the correct relationships between the inputs and predictions. (2) If the parameters do not accurately capture these dependencies, the annealing process will instead yield results that align with the current model's expectations, denoted by $\langle \sigma_i \sigma_j \rangle_{\text{model}}$ and $\langle \sigma_i^2 \rangle_{\text{model}}$. This outcome indicates that the model's parameters require further optimization to better represent the underlying dependencies. Regardless of the parameter accuracy, both scenarios correspond to the equilibrium state of the dynamical system.

## 3.2 HARDWARE AUGMENTATION

The physical realization of this dynamical system is achieved by mapping the node values to the voltages applied on nano-scale capacitors $C$, and modeling the Hamiltonian parameters $J$ and $h$ as the conductance of resistors. More specifically, referring to Fig. 4, the value of the node $\sigma_i$ corresponds to the voltage $V_i$, the effective conductance of the coupling between node $\sigma_i$ and node $\sigma_j$ is $J_{ij}$ (yellow blocks). The effective conductance of the added resistor for node $\sigma_i$ is $2h_i$, which is embedded in the node (green blocks). This mapping enables the construction of the dynamical system using a mesh of programmable resistors, which are interconnected and span across all nodes. By exploiting the intrinsic dynamics (Eq. 12) of this resistor-capacitor network, the natural annealing process can be physically implemented, allowing for rapid convergence toward the equilibrium state that corresponds to the desired results.

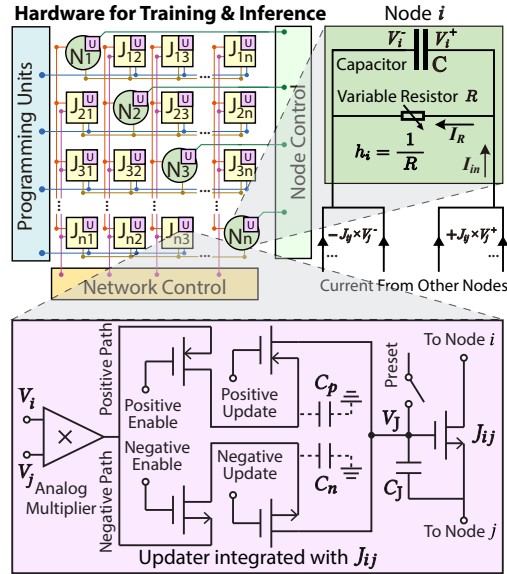

Figure 4: Redesigned InstaTrain hardware.

$$C\frac{dV_i}{dt} = -\frac{\partial \mathcal{H}}{\partial V_i} = \sum_{j \neq i}(J_{ij} + J_{ji})V_j - 2h_iV_i = I_{in} - I_R. \tag{12}$$

After implementing the natural annealing process in this electronic system, we need to further make the system self-trainable. This requires integrating update modules compared to the original design. In particular, the update modules take the values of nodes $\sigma_i$ and $\sigma_j$ as input, compute $V_iV_j$, and update the voltage $V_J$ applied to the capacitor $C_J$. The programmable parameter $J_{ij}$ is then updated according to the value of $V_J$. As depicted in Fig. 4, we embed the update modules (purple blocks) into coupling units (for updating $J_{ij}$) and nodes (for updating $h_i$). The detailed steps are:

1. Initialize $J_{ij}$ and $h_i$ through preset, giving $V_J$ an initial value.
2. Initialize $V_i$ and $V_j$. Set known ground truth values for input nodes, and arbitrary values for remaining nodes.
3. Perform natural annealing to get the updated voltages for unclamped nodes.
4. Obtain $\langle V_iV_j \rangle_{\text{model}}$ using the analog multiplier, storing the results as voltages in capacitors.
5. Load the ground truths for unclamped nodes.
6. Obtain $\langle V_iV_j \rangle_{\text{data}}$ using the analog multiplier, storing the results as voltages in capacitors.
7. Based on the voltage difference between $\langle V_iV_j \rangle_{\text{model}}$ and $\langle V_iV_j \rangle_{\text{data}}$, the positive path is enabled if the former is larger; otherwise, the negative path is enabled.
8. The difference between the two voltages is used to adjust $V_J$, thereby updating $J_{ij}$.
9. Repeat steps (2)-(8) for the next iteration.

Through these steps, the entire training process is transformed into an iterative natural annealing process within the dynamical system, enabling ultra-fast training for highly dynamic applications.

## 4 EVALUATION

### 4.1 EXPERIMENTAL SETUP

**Datasets.** We evaluate InstaTrain on five high-frequency datasets, each providing 100 samples per second. *Carbon-Oxide* consists of sampled time series data collected from a gas delivery platform facility, capturing readings from chemical sensors exposed to varying concentrations of carbon oxide and ethylene mixtures (Fonollosa et al., 2015b). Similarly, *Methane* includes sampled data from chemical sensors exposed to mixtures of methane and ethylene at varying concentration levels (Fonollosa et al., 2015b). *Stock* contains sampled stock data of S&P-500 (Nasdaq). *Ammonia* includes sampled time series recordings from a chemical detection platform, featuring data from 72 metal-oxide sensors across six different locations, all maintained under consistent wind speed and operating temperatures (Fonollosa et al., 2015a). *Toluene* comprises sampled time series recordings from 72 sensors at one location, collected under ten varying conditions (two wind speeds and five operating temperatures) from a chemical detection platform (Fonollosa et al., 2015a).

**Baselines.** We evaluate InstaTrain in three scenarios: static scenarios, low-frequency update scenarios, and high-frequency update scenarios.

- Static Scenarios: Models are trained on the first 25% of each dataset and evaluated on the remaining 75%. We compare against Graph Neural Networks (GNNs), Transformer-based time series prediction models, and NPGL (Wu et al., 2024). The GNNs include: GraphWaveNet (Wu et al., 2019), MTGNN (Wu et al., 2020), and MegaCRN (Jiang et al., 2023). The Transformer-based prediction models include: Autoformer (Wu et al., 2021), DLinear (Zeng et al., 2023), iTransformer (Liu et al., 2023a). All methods are implemented following the experimental setups detailed in their respective original papers.

- Low-Frequency Update Scenarios: Building upon the pre-trained static models above, the GNNs, Transformer-based prediction models, NPGL, and InstaTrain are updated as new data become available. In particular, the models are updated once after observing 1,000 snapshots, equivalent to 10 seconds in the real world. After each update, the model is tested on the subsequent 1,000 snapshots.

Table 1: Accuracy comparison using MAE. Lower values indicate better performance. LF / HF: Low / High Frequency. Gray-shaded results indicate "Not Achievable" results due to slow processing.

| | Dataset | Carbon-Oxide | Methane | Stock | Ammonia | Toluene |
|---|---|---|---|---|---|---|
| Static | GWN | 14.40 | 19.34 | 3.34 | 19.35 | 13.26 |
| | MTGNN | 24.47 | 19.31 | 2.85 | 13.43 | 18.74 |
| | MegaCRN | 25.94 | 23.65 | 3.45 | 18.12 | 20.15 |
| | Informer | 14.16 | 19.37 | 2.76 | 13.59 | 13.07 |
| | DLinear | 14.08 | 19.32 | 2.31 | 12.74 | 12.82 |
| | iTransformer | 14.02 | 19.29 | 2.27 | 12.41 | 11.95 |
| | NPGL | 13.90 | 19.22 | 2.01 | 12.15 | 11.43 |
| | InstaTrain | 13.88 | 19.25 | 2.02 | 12.08 | 11.37 |
| LF Update | GWN | 10.28 | 11.84 | 1.85 | 4.72 | 5.82 |
| | MTGNN | 12.51 | 11.57 | 1.70 | 4.95 | 5.19 |
| | MegaCRN | 12.34 | 13.49 | 1.87 | 5.41 | 5.93 |
| | Informer | 9.21 | 10.41 | 1.64 | 4.39 | 5.25 |
| | DLinear | 8.82 | 10.25 | 1.39 | 4.16 | 4.96 |
| | iTransformer | 8.53 | 9.72 | 1.22 | 3.94 | 4.85 |
| | NPGL | 8.25 | 9.26 | 1.18 | 3.81 | 4.68 |
| | InstaTrain | 8.28 | 9.22 | 1.20 | 3.72 | 4.67 |
| HF Update | FSNet | 7.11 | 7.14 | 0.79 | 1.48 | 2.07 |
| | PatchTST | 7.05 | 7.09 | 0.80 | 1.46 | 2.02 |
| | OneNet | 6.93 | 7.11 | 0.77 | 1.42 | 1.93 |
| | InstaTrain | 6.79 | 7.05 | 0.68 | 1.36 | 1.86 |

- High-Frequency Update Scenarios: Similar to the low-frequency setup, but with models allowed to update more frequently to further improve accuracy. Specifically, InstaTrain is updated once every 100 snapshots (equivalent to 1 second in real time), leveraging its rapid adaptation capability. After each update, the model is evaluated on the subsequent 100 snapshots. In this case, we compare against online learning models, including FSNet (Pham et al., 2022), online-adapted PatchTST (Nie et al., 2022) proposed in (Wen et al., 2024), and OneNet (Wen et al., 2024). These online learning models are implemented based on the setup detailed in their original papers.

**Platforms.** We evaluate the accuracy and inference latency of GNNs, Transformer-based models, and online learning models using an NVIDIA A100-40GB GPU. Training latency for NPGL, GNNs, Transformer-based models, and online learning models is also measured on the same GPU. The accuracy and inference latency of NPGL, along with the accuracy, training latency, and inference latency of InstaTrain, are assessed using a CUDA-accelerated Finite Element Analysis (FEA) software simulator implemented based on BRIM (Afoakwa et al., 2021). Furthermore, the Cadence Mixed-Signal Design Environment is employed to evaluate the power consumption of InstaTrain.

## 4.2 MAIN RESULTS

**Accuracy Evaluations.** Table 1 presents accuracy results across five datasets using MAE as the evaluation metric. Results for dynamic scenarios are averaged across test sets. The comparison across different scenarios indicates that high-frequency updates are crucial for achieving better performance. However, due to computational limitations, GNNs, Transformer-based prediction models, and NPGL can only accommodate the low-frequency update schedule (as shown in Fig. 5). The high-frequency update schedule requires completing updates within 1 second, which is infeasible for these methods. Although selected online learning methods also cannot meet high-frequency update requirements, we still calculate their accuracy under the high-frequency setup for the sake of comparison. The results show that InstaTrain outperforms GNNs and Transformer-based prediction models in both static scenarios and low-frequency update scenarios across all datasets, with comparable accuracy versus NPGL. On average, InstaTrain reduces MAE by 14.45% compared to baselines in static scenarios and by 15.14% in low-frequency update scenarios. In high-frequency update scenarios, InstaTrain outperforms online learning methods with a 6.29% MAE reduction.

Notably, when comparing across different update scenarios, InstaTrain with high-frequency updates achieves a substantial 74.17% MAE reduction compared to all static models, and a 49.51% MAE reduction versus all low-frequency update models, underscoring the critical importance of high-frequency model updating. In summary, the limitation of baselines positions InstaTrain as the optimal solution in the cases where data distributions change rapidly and require frequent adaptation.

**Latency Evaluations.** Figure 5 illustrates the update latency of each method across all datasets. InstaTrain achieves microsecond-level ($10^{-4}$ seconds) update latency, significantly outperforming other approaches that operate at the second level. The orange dashed line represents the update latency requirement for low-frequency model updates (10 seconds). All models perform below this threshold, enabling them to realize the accuracy improvements associated with transitioning from static to low-frequency updating. In contrast, the red dashed line indicates the latency requirement for high-frequency model updates (1 second). Under this criterion, only InstaTrain satisfies the requirement, while all other models fail to achieve the necessary speed for further accuracy enhancements from high-frequency updating.

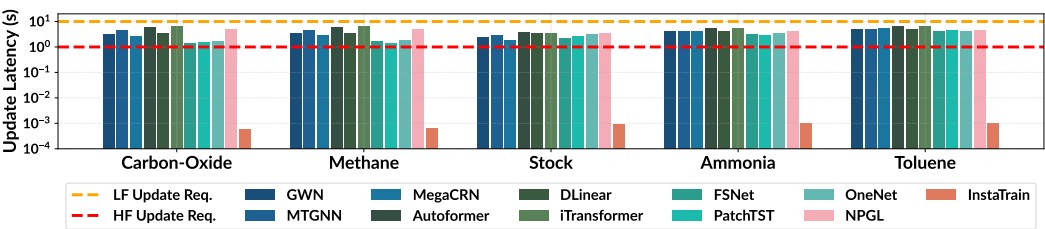

Figure 5: Update latency of methods across datasets.

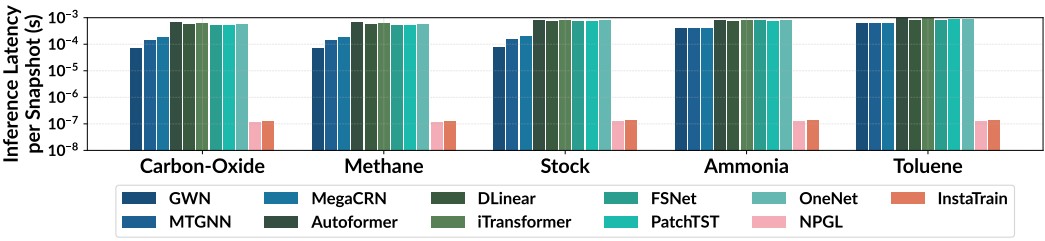

Figure 6: Average inference latency per snapshot.

In addition, the inference latency results in Fig. 6 show that InstaTrain also benefits from the exceptional efficiency brought by the electronic dynamical system, resulting in a similar latency with respect to NPGL. On average, InstaTrain achieves a $\sim$4,000$\times$ speedup in online learning compared to all baseline models across all tasks, while achieving a $\sim$3,000$\times$ speedup in inference compared to all baselines except NPGL. These results underscore the superior computational efficiency of the proposed InstaTrain.

**Power and Energy Consumption.** Table 2 presents a hardware comparison. Compared with the A100 GPU, InstaTrain operates with ultra-low power consumption, requiring approximately 950mW. For a reasonable reference, we assume the average power for the A100 GPU used in this work is 250W. In terms of overall energy consumption, taking

Table 2: Hardware comparison.

| Hardware | Power | Fast Training | Fast Inference |
|---|---|---|---|
| A100 | 250W | No | No |
| NPGL | 500mW | No | Yes |
| InstaTrain | 950mW | **Yes** | **Yes** |

into account the exceptional speedups achieved in training and inference across the selected datasets, InstaTrain achieves more than $10^5$ greater energy efficiency compared to A100 GPUs. Compared to NPGL, despite higher power utilized, InstaTrain supports ultra-fast online training, still resulting in orders-of-magnitude training energy reduction.

**Ablation Study.** Update frequency plays a crucial role in balancing model accuracy and computational efficiency. To investigate its impact, we vary the update interval of InstaTrain from every 50 snapshots to every 1,000 snapshots across all datasets. The results, presented in Table 3, demonstrate that generally, higher-frequency updating achieves better accuracy, as reflected by lower MAE values. This finding further underscores the significance of InstaTrain, which enables extremely high-frequency online updates.

Table 3: Ablation study on update interval.

| Update Interval | Carbon-Oxide | Methane | Stock | Ammonia | Toluene |
|---|---|---|---|---|---|
| 1000 | 8.28 | 9.22 | 1.20 | 3.72 | 4.67 |
| 500 | 7.26 | 8.46 | 0.84 | 2.41 | 3.57 |
| 100 | 6.79 | 7.05 | 0.68 | 1.36 | 1.86 |
| 50 | 6.73 | 7.02 | 0.67 | 1.33 | 1.85 |

## 5 RELATED WORK

**Dynamical Systems in Machine Learning.** Dynamical systems have gained increasing attention in the machine learning community due to their unique properties and their potential to enhance computational efficiency. These systems, which model the evolution of states over time, offer an alternative framework for solving complex optimization problems and machine learning tasks. However, the majority of studies showcasing their potential have been confined to relatively straightforward applications, primarily within the binary domain. For instance, the binary Ising model has been employed to formulate optimization problems (Lucas, 2014), which can be efficiently solved on Ising machines (Mohseni et al., 2022). Additionally, several real-world problems, such as satisfiability

(SAT) problems (Sharma et al., 2023a;b), traffic congestion prediction (Pan et al., 2023), uplink MIMO detection (Singh et al., 2023) and collaborative filtering (Liu et al., 2023b), have also been formulated and addressed using the binary Ising model. While these works have offered valuable insights of using dynamical systems for practical problem solving, they remain constrained by the inherent binary nature of the Ising model, which impedes further progress in real-valued applications in the real world.

Recent efforts, such as those by (Wu et al., 2024), have sought to extend the traditional binary Ising model to accommodate real-valued nodes. Despite this advancement, the practical impact of these extensions has been limited. Firstly, the acceleration achieved by these approaches is confined to the inference phase, leaving the primary bottleneck of training unaddressed. Secondly, the benefits of accelerated inference are diminished if the model is static and cannot be promptly updated, especially in highly dynamic applications where patterns evolve rapidly. In summary, while dynamical systems have proven effective for certain machine learning problems, the slow training process continues to hinder their broader applicability.

**Novel Computing Paradigms.** The limitations of conventional von Neumann architectures have been a bottleneck in modern computing. The separation between processing and memory units, commonly referred to as the "von Neumann bottleneck", limits the performance of CPUs and GPUs. With Moore's Law decelerating and conventional transistor scaling nearing its physical limits, there is an urgent need to explore alternative computing paradigms capable of overcoming these constraints and sustaining performance gains.

In response to these challenges, researchers have investigated several novel computing paradigms. For instance, quantum computing, such as D-Wave (Bunyk et al., 2014), leverages quantum mechanical effects to solve combinatorial optimization problems. Despite their impressive speedups in certain domains, these systems generally require cryogenic environments and are highly specialized, which limits their scalability and broad applicability. Optical computing, such as coherent Ising machines (Yamamoto et al., 2017), exploits the inherent parallelism of light to perform computations at remarkable speeds. However, integration, stability, and reliability issues have hindered their widespread adoption. Inspired by the structure and functionality of biological neural networks, neuromorphic computing has led to the development of systems such as TrueNorth (Akopyan et al., 2015) and Loihi (Davies et al., 2018). Although these systems offer substantial energy efficiency benefits for neural network applications, they often lack the programming flexibility and face challenges in general-purpose computation beyond specific neural algorithms.

Within this diverse landscape, the CMOS compatible Ising machine BRIM (Afoakwa et al., 2021) has emerged as a particularly compelling alternative. BRIM leverages the natural dynamics of electronic circuits to perform computation via natural annealing. Its design is based on analog components such as capacitors and resistors, which allow the system to achieve ultra-fast computation by exploiting the inherent parallelism of physical processes. Moreover, BRIM operates at extremely low power levels—typically on the order of milliwatts—making it an energy-efficient solution. Besides, its compatibility with standard CMOS fabrication techniques facilitates seamless integration into existing semiconductor workflows, which further enhances its practical applicability.

## 6 CONCLUSION

This paper presents InstaTrain, a novel ultra-fast model learning approach for prediction tasks. By transforming the expensive training process into an iterative natural annealing process within a dynamical system, our method enables the model to rapidly adapt to the ever-changing correlations between inputs and predictions, addressing the pressing need for agility and responsiveness in highly dynamic applications. This pioneering approach transcends the limitations of conventional methods and paves the way for a new era of ultra-fast and energy-efficient learning, empowering applications in domains characterized by high data volatility and stringent latency requirements. Further explorations could focus on incorporating advanced online learning strategies that are compatible with the hardware, which might yield better solutions. To highlight, InstaTrain achieves, on average, a $\sim$4,000$\times$ speedup in training supporting microsecond-level model updates and a $\sim 10^5 \times$ reduction in training energy cost, along with lower MAE compared to baselines running on GPUs.

## ACKNOWLEDGMENTS

This work is supported by the U.S. Department of Energy, Office of Science, Office of Advanced Scientific Computing Research, in support of the MEERCAT Microelectronics Science Research Center, under Contract DE-AC05-76RL01830. This work is also supported by NSF under Award No. 2326494.

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
