# OpenReview forum: "InstaTrain: Adaptive Training via Ultra-Fast Natural Annealing within Dynamical Systems"
_ICLR.cc/2025/Conference — ICLR 2025 Poster_

### Official Review · Reviewer_3cwj · 2024-11-03

**Soundness:** 3
**Presentation:** 3
**Contribution:** 3
**Rating:** 6
**Confidence:** 2

**Summary:**

The paper introduces InstaTrain, a novel adaptive learning paradigm designed to operate in highly dynamic environments where data distributions shift rapidly. InstaTrain leverages a natural annealing process on an electronic dynamical system, enabling model training and updates at microsecond intervals. Experimental results across high-frequency datasets showcase substantial improvements in speed, energy efficiency, and prediction accuracy compared to state-of-the-art (SOTA) methods.

**Strengths:**

1. The application of natural annealing as a training mechanism in a dynamical system is novel, addressing limitations in existing adaptive models in terms of speed and energy consumption.

2. InstaTrain demonstrates significant training speedups (4,135×) and energy reductions (105×) while achieving superior accuracy on high-frequency datasets. This supports the paper’s claims of real-time adaptability and efficiency.

3. The integration of parameter update modules within the dynamical system facilitates simultaneous training and inference, a promising approach for handling rapid data shifts.

**Weaknesses:**

1. The model appears to primarily handle binary data (+1, -1). How does the proposed InstaTrain method address continuous data in time series applications?

2. The paper omits several widely used time series datasets, including ETTm1, Electricity, Traffic, Exchange, and Weather.

**Questions:**

see weakness

---

> ### Author Response · Authors · 2024-11-21
> **Response to Reviewer 3cwj**
>
> Thank you for your valuable review. We will address your questions one by one below.
>
> **Explanation of how InstaTrain addresses continuous data.**
>
> We apologize for any confusion. To clarify, the Hamiltonian used in our work is a novel modification proposed by [1], which extends the binary Ising model to support real-valued nodes. Specifically, the binary limitation of the Ising model refers to the failure of naively extending its binary nodes to real values. The Hamiltonian of binary Ising model is $H(\sigma) = -\sum_{i\neq j}^{N}{J_{ij} \sigma_i \sigma_j} - \sum_{i}^{N}{h_i \sigma_i}$. If $\sigma$ are real-valued, they evolve to $\pm\infty$ to pursue the lowest energy state, which is $-\infty$. Even if boundaries are applied to $\sigma$, $\sigma$ are only intercepted along their way to infinity, resulting in polarized nodes and essentially a binary model.
>
> To resolve this, [1] replaces the linear term in the original Hamiltonian with a pure quadratic term, resulting in:
> $H_{rv} = -\sum_{i\neq j}^{N}{J_{ij} x_i x_j} + \sum_{i}^{N}{h_i x_i^2},~x_i\in R.$
> $X=\left ( x_1, x_2, ......, x_N \right )$ denotes the nodes in the dynamical system, $J_{ij}$ represents the relationship between node $i$ and node $j$, and $h_i$ refers to the self-reaction strength and is forced positive. The quadratic term acts as an energy regulator, which prevents the energy from going down to $-\infty$, allowing nodes to be localized in a certain range.
> We have expanded the Background section (Ising-Based Hamiltonian) in the revised manuscript to enhance understanding.
>
> **The reason of excluding several widely used time series datasets.**
>
> Thanks for your valuable comments. We are currently running and comparing our method on widely used datasets and will provide the results soon. To explain why we previously did not include them, we would like to clarify that our focus is on proposing a novel online learning solution for highly dynamic environments through natural annealing within dynamical systems. Therefore, we evaluate on high-frequency datasets with rapidly evolving distributions (0.01 seconds per sample).  We also want to highlight some improvements we currently finished to ensure a more comprehensive evaluation. Specifically, we add two more challenging high-frequency datasets (Ammonia and Toluene), both collected at a sampling rate of 100 Hz. We also add three new Transformer-based baselines (Autoformer [2], DLinear [3], iTransformer [4]). Our evaluation now includes a total of eleven models across five datasets and three scenarios. The Evaluation section has been updated accordingly. We attach the accuracy comparison table below for your reference.
>
> **Table:** Accuracy comparison across datasets. LF / HF: Low / High Frequency. Online learning methods only have HF. In HF update, the results from the baselines are “Not Achievable” due to slow training.
> | Category      | Dataset      | Carbon-Oxide | Methane | Stock | Ammonia | Toluene |
> |---------------|--------------|--------------|---------|-------|---------|---------|
> | **Static**    | Best GNN     | 14.40        | 19.31   | 2.85  | 13.43   | 13.26   |
> |               | Best Trans   | 14.02        | 19.29   | 2.27  | 12.41   | 11.95   |
> |               | NPGL         | 13.90        | 19.22   | 2.01  | 12.15   | 11.43   |
> |               | InstaTrain   | 13.88        | 19.25   | 2.02  | 12.08   | 11.37   |
> | **LF Update** | Best GNN     | 10.28        | 11.57   | 1.70  | 4.72    | 5.19    |
> |               | Best Trans   | 8.53         | 9.72    | 1.22  | 3.94    | 4.85    |
> |               | NPGL         | 8.25         | 9.26    | 1.18  | 3.81    | 4.68    |
> |               | InstaTrain   | 8.28         | 9.22    | 1.20  | 3.72    | 4.67    |
> | **HF Update** | Best GNN     | *7.16*       | *7.36*  | *0.80* | *1.62*  | *2.11*  |
> |               | Best Trans   | *7.12*       | *7.25*  | *0.73* | *1.45*  | *1.94*  |
> |               | FSNet        | *7.11*       | *7.14*  | *0.79* | *1.48*  | *2.07*  |
> |               | PatchTST     | *7.05*       | *7.09*  | *0.80* | *1.46*  | *2.02*  |
> |               | OneNet       | *6.93*       | *7.11*  | *0.77* | *1.42*  | *1.93*  |
> |               | NPGL         | *6.81*       | *7.08*  | *0.68* | *1.39*  | *1.90*  |
> |               | InstaTrain   | 6.79         | 7.05    | 0.68  | 1.36    | 1.86    |
>
> *Note: Results marked in italics represent “Not Achievable” due to slow training.*
>
> [1] Wu, C., Song, R., Liu, C., Yang, Y., Li, A., Huang, M., & Geng, T.. Extending Power of Nature from Binary to Real-Valued Graph Learning in Real World.
>
> [2] Wu, H., Xu, J., Wang, J., & Long, M.. Autoformer: Decomposition transformers with auto-correlation for long-term series forecasting.
>
> [3] Zeng, A., Chen, M., Zhang, L., & Xu, Q.. Are transformers effective for time series forecasting?
>
> [4] Liu, Y., Hu, T., Zhang, H., Wu, H., Wang, S., Ma, L., & Long, M.. iTransformer: Inverted Transformers Are Effective for Time Series Forecasting.

---

> > ### Author Response · Authors · 2024-11-23
> > **Response to Reviewer 3cwj**
> >
> > **Results on Widely Used Time Series Datasets.**
> >
> > Thank you once again for your valuable suggestion to incorporate widely used time series datasets into our evaluation. We have now completed experiments on the ETTm1, Electricity, Traffic, and Weather datasets using several SOTA models, including MegaCRN (a SOTA GNN) [1], iTransformer (a SOTA Transformer-based model) [2], NPGL [3], three SOTA online learning baselines (FSNet [4], the online learning version of PatchTST [5], OneNet [5]), and the proposed InstaTrain. Given that the highest frequency of these datasets is 10 minutes per sample, all baselines meet the high-frequency update schedule. Therefore, we evaluate the models in two scenarios: the static scenario and the high-frequency (HF) update scenario. The results are shown in the table below.
> >
> > Table: MAE comparison across datasets.
> > |         | Model         | ETTm1 | Electricity | Traffic | Weather |
> > |---------|---------------|-------|-------------|---------|---------|
> > | Static  | MegaCRN       | 0.416 | 0.420        | 0.397   | 0.301   |
> > |         | iTransformer  | 0.372 | 0.379       | 0.329   | 0.246   |
> > |         | NPGL          | 0.367 | 0.375       | 0.323   | 0.235   |
> > |         | InstaTrain    | 0.369 | 0.375       | 0.320   | 0.231   |
> > | HF      | MegaCRN       | 0.257 | 0.309       | 0.285   | 0.224   |
> > |         | iTransformer  | 0.213 | 0.290        | 0.268   | 0.221   |
> > |         | FSNet         | 0.191 | 0.472       | 0.253   | 0.216   |
> > |         | PatchTST      | 0.186 | 0.224       | 0.241   | 0.200     |
> > |         | OneNet        | 0.187 | 0.254       | 0.243   | 0.201   |
> > |         | NPGL          | 0.174 | 0.216       | 0.235   | 0.194   |
> > |         | InstaTrain    | 0.172 | 0.214       | 0.231   | 0.196   |
> >
> > In the static scenario, iTransformer, NPGL, and InstaTrain show competitive performance. Specifically, InstaTrain outperforms NPGL on the Traffic and Weather datasets while maintaining comparable performance on ETTm1 and Electricity datasets. In the high-frequency update scenario, all models show improved performance compared to the static scenario. InstaTrain achieves the lowest MAE across all datasets, outperforming baselines. Notably, FSNet exhibits a higher MAE on the Electricity dataset in the HF scenario, probably due to its lack of robustness with this dataset. The consistent superior performance of InstaTrain in the HF scenario underscores its effectiveness in leveraging high-frequency updates to enhance performance.
> >
> >
> > **Sincerely Seeking Advice.**
> >
> > Dear Reviewer 3cwj, we deeply appreciate the time and effort you have dedicated to reviewing our work and for providing constructive feedback. As the discussion period approaches its ending, we kindly ask if you could share any additional thoughts or suggestions regarding our clarifications. We would like to ensure that we have adequately addressed your concerns, and any further insights from you would be immensely valuable for us to improve.
> >
> > Thank you once again for your thoughtful engagement and support throughout this process.
> >
> > Best regards,
> >
> > The Authors
> >
> > [1] Renhe Jiang, Zhaonan Wang, Jiawei Yong, Puneet Jeph, Quanjun Chen, Yasumasa Kobayashi, Xuan Song, Shintaro Fukushima, and Toyotaro Suzumura. Spatio-temporal meta-graph learning for traffic forecasting. In Proceedings of the AAAI Conference on Artificial Intelligence.
> >
> > [2] Liu, Y., Hu, T., Zhang, H., Wu, H., Wang, S., Ma, L., & Long, M. iTransformer: Inverted Transformers Are Effective for Time Series Forecasting. In The Twelfth International Conference on Learning Representations.
> >
> > [3] Wu, C., Song, R., Liu, C., Yang, Y., Li, A., Huang, M., & Geng, T. Extending Power of Nature from Binary to Real-Valued Graph Learning in Real World. In The Twelfth International Conference on Learning Representations.
> >
> > [4] Pham, Q., Liu, C., Sahoo, D., & Hoi, S. Learning Fast and Slow for Online Time Series Forecasting. In The Eleventh International Conference on Learning Representations.
> >
> > [5] Wen, Q., Chen, W., Sun, L., Zhang, Z., Wang, L., Jin, R., & Tan, T. Onenet: Enhancing time series forecasting models under concept drift by online ensembling. Advances in Neural Information Processing Systems.

---

> > > ### Comment · Reviewer_3cwj · 2024-11-25
> > >
> > > Thank you for your response, which addressed my concerns and I am willing to improve my score.

---

> ### Author Response · Authors · 2024-11-25
>
> Dear Reviewer 3cwj,
>
> Thank you for your thoughtful feedback and for taking the time to review our response. We sincerely appreciate your acceptance of our work. Your constructive insights have been invaluable in improving the quality of the manuscript. Please do not hesitate to let us know if there are any remaining concerns or additional details that we can address to further improve the manuscript.
>
> Thank you once again for your thorough and valuable review.
>
> Best regards,
>
> The Authors

---

### Official Review · Reviewer_hPWn · 2024-11-03

**Soundness:** 2
**Presentation:** 2
**Contribution:** 2
**Rating:** 5
**Confidence:** 3

**Summary:**

This paper introduces InstaTrain, a novel approach for ultra-fast model training in highly dynamic environments where data distributions can change within microseconds. Traditional models, including online learning approaches, struggle to adapt quickly enough to such rapid changes. The authors transform the typically slow training process into an ultra-fast "natural annealing" process implemented on an electronic dynamical system. Their key contribution is twofold: they develop a new training algorithm called Iterative Natural Annealing based Training (INAT) that reformulates model training as an energy minimization problem, and they augment existing electronic hardware with parameter update modules to enable both training and inference on the same system. Evaluated on three high-frequency datasets including stock market and chemical sensor data, InstaTrain achieves dramatic improvements over state-of-the-art methods, demonstrating orders of magnitude faster training and inference speeds, significantly lower energy consumption, and better prediction accuracy. The work effectively bridges physics-inspired computing with machine learning to address the challenge of rapid distribution shifts in real-world applications.

**Strengths:**

Originality: The paper creatively combines statistical physics with machine learning in a novel way, transforming traditional model training into a physical annealing process. While both Ising models and online learning existed previously, extending electronic hardware to support microsecond-level training and inference together represents a genuine innovation. The work effectively removes the speed limitations that have constrained previous approaches to dynamic model adaptation.
Quality: The technical work is rigorous, with thorough mathematical foundations and experimental validation. The evaluation compares against multiple baselines across several high-frequency datasets, demonstrating improvements in speed, energy efficiency, and accuracy. The inclusion of ablation studies further strengthens the empirical results.
Clarity: The paper effectively communicates complex concepts through clear writing.
Significance: The work has substantial practical impact by enabling model adaptation at previously impossible speeds while reducing energy costs by orders of magnitude. It also opens promising research directions by demonstrating how physics-based computing can be effectively combined with machine learning to overcome fundamental limitations of traditional approaches.

**Weaknesses:**

- The baseline comparison is incomplete. The paper should compare to more regular time series prediction models.
- The presentation of the results in bar plot is not ideal. Tables of numbers are much better.

**Questions:**

Can you add more comparison results in the form of tables?

---

> ### Author Response · Authors · 2024-11-21
> **Response to Reviewer hPWn**
>
> Thank you for the insightful reviews. We will answer your questions one by one in the following.
>
> **Compare with other time series prediction models**
>
> Thanks for your valuable suggestion. In response, we have added three widely used Transformer-based models in time series prediction: Autoformer [1], DLinear [2], iTransformer [3]. Additionally, we have also included two more challenging datasets: Ammonia and Toluene, to provide a more comprehensive comparison. Ammonia contains time series recordings from 72 metal-oxide sensors across six different locations under consistent wind speed and operating temperatures, resulting in a (12,755 × 432) array. Toluene includes time series recordings from 72 sensors at one location under ten varying conditions (two wind speeds and five operating temperatures), forming a (12,854 × 720) array.
>
> Our evaluation now includes a total of eleven models across five datasets and three scenarios: static, low-frequency updates, and high-frequency updates. The evaluation section has been updated accordingly. The results are presented in the Table 1, Figure 5, and Figure 6 of the revised manuscript. We have attached the accuracy comparison table below for your reference. Here, we present the performance of the best-performing GNN and Transformer-based model (Best Trans), a complete version of this table is provided in Table 5 in the Appendix of the revised manuscript.
>
> **Table:** Accuracy comparison across datasets. LF / HF: Low / High Frequency. Online learning methods only have HF. In HF update, the results from the baselines are “Not Achievable” due to slow training.
> | Category      | Dataset      | Carbon-Oxide | Methane | Stock | Ammonia | Toluene |
> |---------------|--------------|--------------|---------|-------|---------|---------|
> | **Static**    | Best GNN     | 14.40        | 19.31   | 2.85  | 13.43   | 13.26   |
> |               | Best Trans   | 14.02        | 19.29   | 2.27  | 12.41   | 11.95   |
> |               | NPGL         | 13.90        | 19.22   | 2.01  | 12.15   | 11.43   |
> |               | InstaTrain   | 13.88        | 19.25   | 2.02  | 12.08   | 11.37   |
> | **LF Update** | Best GNN     | 10.28        | 11.57   | 1.70  | 4.72    | 5.19    |
> |               | Best Trans   | 8.53         | 9.72    | 1.22  | 3.94    | 4.85    |
> |               | NPGL         | 8.25         | 9.26    | 1.18  | 3.81    | 4.68    |
> |               | InstaTrain   | 8.28         | 9.22    | 1.20  | 3.72    | 4.67    |
> | **HF Update** | Best GNN     | *7.16*       | *7.36*  | *0.80* | *1.62*  | *2.11*  |
> |               | Best Trans   | *7.12*       | *7.25*  | *0.73* | *1.45*  | *1.94*  |
> |               | FSNet        | *7.11*       | *7.14*  | *0.79* | *1.48*  | *2.07*  |
> |               | PatchTST     | *7.05*       | *7.09*  | *0.80* | *1.46*  | *2.02*  |
> |               | OneNet       | *6.93*       | *7.11*  | *0.77* | *1.42*  | *1.93*  |
> |               | NPGL         | *6.81*       | *7.08*  | *0.68* | *1.39*  | *1.90*  |
> |               | InstaTrain   | 6.79         | 7.05    | 0.68  | 1.36    | 1.86    |
>
> *Note: Results marked in italics represent “Not Achievable” due to slow training.*
>
> **Using Tables to present results**
>
> Thanks for your valuable suggestion. In the revised manuscript, we have incorporated tables to present our results more clearly and effectively. Specifically, Tables 1, 2, and 3 have been included in the main text to highlight the key findings, while Tables 4, 5, and 6 have been added to the Appendix to provide supplementary details. We greatly appreciate your insightful feedback, which has helped enhance the clarity of our work.
>
> [1] Wu, H., Xu, J., Wang, J., & Long, M. (2021). Autoformer: Decomposition transformers with auto-correlation for long-term series forecasting. Advances in neural information processing systems, 34, 22419-22430.
>
> [2] Zeng, A., Chen, M., Zhang, L., & Xu, Q. (2023). Are transformers effective for time series forecasting?. In Proceedings of the AAAI conference on artificial intelligence (Vol. 37, No. 9, pp. 11121-11128).
>
> [3] Liu, Y., Hu, T., Zhang, H., Wu, H., Wang, S., Ma, L., & Long, M. (2024). iTransformer: Inverted Transformers Are Effective for Time Series Forecasting. In The Twelfth International Conference on Learning Representations.

---

> > ### Author Response · Authors · 2024-11-23
> > **Sincerely Seeking Advice**
> >
> > Dear Reviewer hPWn,
> >
> > We deeply appreciate the time and effort you have dedicated to reviewing our work and for providing constructive feedback. As the discussion period approaches its ending, we kindly ask if you could share any additional thoughts or suggestions regarding our clarifications. We would like to ensure that we have adequately addressed your concerns, and any further insights from you would be immensely valuable for us to improve.
> >
> > Thank you once again for your thoughtful engagement and support throughout this process.
> >
> > Best regards,
> >
> > The Authors

---

> > > ### Author Response · Authors · 2024-11-27
> > > **Gratitude and Further Suggestions**
> > >
> > > Dear Reviewer hPWn,
> > >
> > > We wish to express our deepest gratitude for your insightful comments and valuable suggestions, which have significantly enhanced the quality of our paper. As we are preparing to submit a new revision, we would be grateful if you could kindly provide any additional feedback or suggestions you may have. Addressing your concerns is of the utmost importance to us, and any further insights from you would be immensely beneficial in improving our work.
> > >
> > > Once again, we sincerely thank you for your thoughtful engagement and support throughout this process.
> > >
> > > Best regards,
> > >
> > > The Authors

---

> > > > ### Author Response · Authors · 2024-11-28
> > > >
> > > > Dear Reviewer hPWn,
> > > >
> > > > We would like to inform you that we have uploaded a new revision of our paper. Your insightful suggestions have significantly improved our work, and we are truly grateful for your feedback. We would greatly appreciate any additional feedback or suggestions you may have, as addressing your concerns is of the utmost importance to us. We sincerely thank you for your thoughtful support.
> > > >
> > > > Best regards,
> > > >
> > > > The Authors

---

> > > > > ### Author Response · Authors · 2024-12-01
> > > > > **Kindly Requesting Follow-Up**
> > > > >
> > > > > Dear Reviewer hPWn,
> > > > >
> > > > > Thank you for taking the time to review our work. Your insightful feedback is invaluable to us, and we are sincerely grateful for your thoughtful contributions.
> > > > >
> > > > > Given that the discussion phase is nearing its conclusion, we would be most grateful for the opportunity to engage in further dialogue with you before it ends. Our goal is to ensure that our responses address your concerns effectively, and to explore any additional questions or comments you may have.
> > > > >
> > > > > We greatly appreciate your contributions to this process and look forward to your input.
> > > > >
> > > > > Best regards,
> > > > >
> > > > > The Authors

---

> ### Comment · Reviewer_hPWn · 2024-12-03
>
> Thank you for all the responses and the additional experiments. I think it's also necessary to demonstrate the model's performance on those commonly used time series datasets although they might not be of high frequency to better compare with other time series models. I will keep my scores.

---

> ### Author Response · Authors · 2024-12-03
>
> Dear Reviewer hPWn,
>
> We sincerely appreciate your feedback and valuable suggestion to include widely used time series datasets in our evaluation. To clarify, following Reviewer 3cwj's suggestion, we incorporated experiments and results on commonly used time series datasets in our revision. These additions can be found in Appendix A.3 and Table 6. We deeply apologize for not highlighting these additional experiments in our earlier response.
>
> For your convenience, we have also included the results in Table 6 below.
>
> We added experiments on the ETTm1, Electricity, Traffic, and Weather datasets using several SOTA models, including MegaCRN (a SOTA GNN) [1], iTransformer (a SOTA Transformer-based model) [2], NPGL [3], three SOTA online learning baselines (FSNet [4], the online learning version of PatchTST [5], OneNet [5]), and the proposed InstaTrain. Given that the highest frequency of these datasets is 10 minutes per sample, all baselines meet the high-frequency update schedule. Therefore, we evaluate the models in two scenarios: the static scenario and the high-frequency (HF) update scenario. The results are shown in the table below.
>
> Table: MAE comparison across datasets.
> |         | Model         | ETTm1 | Electricity | Traffic | Weather |
> |---------|---------------|-------|-------------|---------|---------|
> | Static  | MegaCRN       | 0.416 | 0.420        | 0.397   | 0.301   |
> |         | iTransformer  | 0.372 | 0.379       | 0.329   | 0.246   |
> |         | NPGL          | 0.367 | 0.375       | 0.323   | 0.235   |
> |         | InstaTrain    | 0.369 | 0.375       | 0.320   | 0.231   |
> | HF      | MegaCRN       | 0.257 | 0.309       | 0.285   | 0.224   |
> |         | iTransformer  | 0.213 | 0.290        | 0.268   | 0.221   |
> |         | FSNet         | 0.191 | 0.472       | 0.253   | 0.216   |
> |         | PatchTST      | 0.186 | 0.224       | 0.241   | 0.200     |
> |         | OneNet        | 0.187 | 0.254       | 0.243   | 0.201   |
> |         | NPGL          | 0.174 | 0.216       | 0.235   | 0.194   |
> |         | InstaTrain    | 0.172 | 0.214       | 0.231   | 0.196   |
>
> In the static scenario, iTransformer, NPGL, and InstaTrain show competitive performance. Specifically, InstaTrain outperforms NPGL on the Traffic and Weather datasets while maintaining comparable performance on ETTm1 and Electricity datasets. In the high-frequency update scenario, all models show improved performance compared to the static scenario. InstaTrain achieves the lowest MAE across all datasets, outperforming baselines. Notably, FSNet exhibits a higher MAE on the Electricity dataset in the HF scenario, probably due to its lack of robustness with this dataset. The consistent superior performance of InstaTrain in the HF scenario underscores its effectiveness in leveraging model updates to enhance performance.
>
> Thank you once again for your thoughtful and constructive comments and reviews.
>
> Best regards,
>
> The Authors
>
> [1] Renhe Jiang, Zhaonan Wang, Jiawei Yong, Puneet Jeph, Quanjun Chen, Yasumasa Kobayashi, Xuan Song, Shintaro Fukushima, and Toyotaro Suzumura. Spatio-temporal meta-graph learning for traffic forecasting. In Proceedings of the AAAI Conference on Artificial Intelligence.
>
> [2] Liu, Y., Hu, T., Zhang, H., Wu, H., Wang, S., Ma, L., & Long, M. iTransformer: Inverted Transformers Are Effective for Time Series Forecasting. In The Twelfth International Conference on Learning Representations.
>
> [3] Wu, C., Song, R., Liu, C., Yang, Y., Li, A., Huang, M., & Geng, T. Extending Power of Nature from Binary to Real-Valued Graph Learning in Real World. In The Twelfth International Conference on Learning Representations.
>
> [4] Pham, Q., Liu, C., Sahoo, D., & Hoi, S. Learning Fast and Slow for Online Time Series Forecasting. In The Eleventh International Conference on Learning Representations.
>
> [5] Wen, Q., Chen, W., Sun, L., Zhang, Z., Wang, L., Jin, R., & Tan, T. Onenet: Enhancing time series forecasting models under concept drift by online ensembling. Advances in Neural Information Processing Systems.

---

### Official Review · Reviewer_2k7C · 2024-11-04

**Soundness:** 3
**Presentation:** 3
**Contribution:** 3
**Rating:** 6
**Confidence:** 3

**Summary:**

This paper presents a novel approach that transforms traditional model training into an iterative natural annealing process within an electronic dynamical system. The proposed approach operates through Iterative Natural Annealing-based Training (INAT), which performs training directly on an electronic dynamical system rather than on conventional processors. The key innovation lies in extending existing electronic dynamical system hardware from inference-only to support both training and inference on the same hardware. The authors claim significant improvements in training speed, energy efficiency, and prediction accuracy compared to state-of-the-art methods, particularly for time-series prediction tasks with rapidly evolving data distributions.

**Strengths:**

- The novel transformation of traditional training into a natural annealing process is interesting.
- Hardware-algorithm co-design that extends existing electronic dynamical systems.
- Clear theoretical foundation.
- Well-structured presentation with clear problem motivation.

**Weaknesses:**

1. Some concerns about the experimental datasets. As far as I know, the three datasets used are time series data with relatively low-dimension. There is no discussion of scalability to larger, more complex datasets. How would the system perform on higher-dimensional data or more complex tasks? Are there theoretical or practical limitations?

2. The experimental results of the proposed IntraTrain approach are based on CUDA-based software simulator rather than actual hardware implementation. Although the author claimed that the real hardware has been manufactured in the reproducibility part.

3. In Table 1, the power consumption of InstaTrain (950mW) is actually higher than baseline BRIM (250mW) and NP-GL (260mW), could you provide detailed calculations for the claimed energy reduction?

4. The paper compares performance with "not achievable" baseline results for high-frequency scenarios. Could you explain the methodology behind these comparisons since they are "not achievable"?

I'd be happy to raise the score if the author could address my concerns.

**Questions:**

See the weaknesses part.

---

> ### Author Response · Authors · 2024-11-21
> **Response to Reviewer 2k7C (Part 1)**
>
> Thanks for your valuable reviews. We will address your concerns one by one in the following.
>
> **How InstaTrain would perform with higher-dimensional data, as well as any theoretical or practical limitations?**
>
> Thanks for your valuable question. To evaluate our model's performance on higher-dimensional data, we have conducted experiments on two additional high-frequency data from a chemical detection platform, both collected at a sampling rate of 100 Hz:
>
> 1. Ammonia contains time series recordings from 72 metal-oxide sensors across six different locations under consistent wind speed and operating temperatures, resulting in a (12,755 × 432) array.
>
> 2. Toluene includes time series recordings from 72 sensors at one location under ten varying conditions (two wind speeds and five operating temperatures), forming a (12,854 × 720) array.
>
> In addition, to provide a more comprehensive comparison, we have also added three new Transformer-based baselines (Autoformer [1], DLinear [2], iTransformer [3]). Our evaluation now includes a total of eleven models across five datasets and three scenarios: static, low-frequency updates, and high-frequency updates. The Evaluation section has been updated accordingly. The results are presented in the Table 1, Figure 5, and Figure 6 of the revised manuscript. We have attached the accuracy comparison table below for your reference. Here, we present the performance of the best-performing GNN and Transformer-based model (Best Trans), a complete version of this table is provided in Table 5 in the Appendix of the revised manuscript.
>
> There are no limitations for the dynamical system processor to scale to higher-dimensional data. It has been demonstrated that dynamical system processors inherently support spatial-temporal co-annealing capabilities [4]. Spatial co-annealing allows multiple dynamical system processors to collaboratively find the lowest energy of a joint energy landscape without slowing down the process. Temporal co-annealing ensures that the processor can efficiently find the lowest energy state by continuously searching different sub-regions of the entire energy landscape over time until convergence.
>
> **Table:** Accuracy comparison across datasets. LF / HF: Low / High Frequency. Online learning methods only have HF. In HF update, the results from the baselines are “Not Achievable” due to slow training.
> | Category      | Dataset      | Carbon-Oxide | Methane | Stock | Ammonia | Toluene |
> |---------------|--------------|--------------|---------|-------|---------|---------|
> | **Static**    | Best GNN     | 14.40        | 19.31   | 2.85  | 13.43   | 13.26   |
> |               | Best Trans   | 14.02        | 19.29   | 2.27  | 12.41   | 11.95   |
> |               | NPGL         | 13.90        | 19.22   | 2.01  | 12.15   | 11.43   |
> |               | InstaTrain   | 13.88        | 19.25   | 2.02  | 12.08   | 11.37   |
> | **LF Update** | Best GNN     | 10.28        | 11.57   | 1.70  | 4.72    | 5.19    |
> |               | Best Trans   | 8.53         | 9.72    | 1.22  | 3.94    | 4.85    |
> |               | NPGL         | 8.25         | 9.26    | 1.18  | 3.81    | 4.68    |
> |               | InstaTrain   | 8.28         | 9.22    | 1.20  | 3.72    | 4.67    |
> | **HF Update** | Best GNN     | *7.16*       | *7.36*  | *0.80* | *1.62*  | *2.11*  |
> |               | Best Trans   | *7.12*       | *7.25*  | *0.73* | *1.45*  | *1.94*  |
> |               | FSNet        | *7.11*       | *7.14*  | *0.79* | *1.48*  | *2.07*  |
> |               | PatchTST     | *7.05*       | *7.09*  | *0.80* | *1.46*  | *2.02*  |
> |               | OneNet       | *6.93*       | *7.11*  | *0.77* | *1.42*  | *1.93*  |
> |               | NPGL         | *6.81*       | *7.08*  | *0.68* | *1.39*  | *1.90*  |
> |               | InstaTrain   | 6.79         | 7.05    | 0.68  | 1.36    | 1.86    |
>
> *Note: Results marked in italics represent “Not Achievable” due to slow training.*
>
>
> [1] Wu, H., Xu, J., Wang, J., & Long, M. (2021). Autoformer: Decomposition transformers with auto-correlation for long-term series forecasting. Advances in neural information processing systems, 34, 22419-22430.
>
> [2] Zeng, A., Chen, M., Zhang, L., & Xu, Q. (2023). Are transformers effective for time series forecasting?. In Proceedings of the AAAI conference on artificial intelligence (Vol. 37, No. 9, pp. 11121-11128).
>
> [3] Liu, Y., Hu, T., Zhang, H., Wu, H., Wang, S., Ma, L., & Long, M. (2024). iTransformer: Inverted Transformers Are Effective for Time Series Forecasting. In The Twelfth International Conference on Learning Representations.
>
> [4] Song, R., Wu, C., Liu, C., Li, A., Huang, M., & Geng, T. (2024). DS-GL: Advancing graph learning via harnessing nature's power within scalable dynamical systems. In Proceedings of the 51st IEEE/ACM International Symposium on Computer Architecture.

---

> > ### Author Response · Authors · 2024-11-21
> > **Response to Reviewer 2k7C (Part 2)**
> >
> > **Experimental results are based on the simulator rather than actual hardware implementation.**
> >
> > Thanks for your valuable comment. Our Finite Element Analysis simulator, adapted from [5], is a SOTA simulation approach for circuit-level simulation, accurately simulating voltage and current evolution with 100-picosecond time steps. This simulator has been validated to produce accurate results compared to the manufactured hardware on small-scale problems. Currently, the accessible and well-tested hardware enables addressing small-scale problems, and the implementation of large-scale problems is still undergoing testing. Therefore, all our experimental results are based on the simulator, facilitating easy verification and future research in the field. We have revised the Reproducibility section to clarify this. Thank you again for your comment.
> >
> > **Detailed calculations for the claimed energy reduction.**
> >
> > We apologize for any confusion. The claimed energy reduction is calculated based on both power consumption and training speedup. Specifically, InstaTrain achieves an average training speedup of over $10^3$ times compared to all baseline models across various datasets. InstaTrain operates at a power consumption of 950mW, while an NVIDIA A100 GPU operates at a power of more than 100W. Therefore, the energy reduction is more than $10^3 * 10^2 = 10^5$. Although InstaTrain’s power is slightly higher than that of BRIM and NPGL, it enables ultra-fast learning within dynamical systems, which is not achievable for the other two methods.
> >
> > **Explanation for the not achievable results.**
> >
> > We are sorry for any confusion and appreciate the opportunity to clarity the meaning of “Not Achievable” results. To provide a comprehensive comparison, we evaluate InstaTrain alongside baseline models in three scenarios: static, low-frequency update, and high-frequency update. In the high-frequency update scenario, all models are set to update once every 100 snapshots, equivalent to a 1-second real-time interval. This setup requires models to complete parameter updates within 1 second to be ready for the next cycle. However, as shown in Figure 5, the training latency per snapshot for all baseline models exceeds 0.01 seconds, resulting in a cumulative training time of over 1 second for 100 snapshots. This delay means that baseline models cannot keep up with the high-frequency update schedule, making real-time adaptation “Not Achievable” for them in this scenario.
> > However, for the sake of comparison, we still calculate the accuracy of these baseline models under the high-frequency setup, assuming they could meet the update interval. This serves two purposes: (1) to emphasize the importance of high-frequency updates for improved performance on high-frequency datasets and (2) to provide a comprehensive comparison of accuracy across models.
> > We hope this clarifies the “Not Achievable” results and we have revised the manuscript to include this explanation for clarity.
> >
> >
> > [5] Richard Afoakwa, Yiqiao Zhang, Uday Kumar Reddy Vengalam, Zeljko Ignjatovic, and Michael Huang. Brim: Bistable resistively-coupled ising machine. In 2021 IEEE International Symposium on High-Performance Computer Architecture (HPCA), pp. 749–760. IEEE, 2021.

---

> > > ### Author Response · Authors · 2024-11-23
> > > **Sincerely Seeking Advice**
> > >
> > > Dear Reviewer 2k7C,
> > >
> > > We deeply appreciate the time and effort you have dedicated to reviewing our work and for providing constructive feedback. As the discussion period approaches its ending, we kindly ask if you could share any additional thoughts or suggestions regarding our clarifications. We would like to ensure that we have adequately addressed your concerns, and any further insights from you would be immensely valuable for us to improve.
> > >
> > > Thank you once again for your thoughtful engagement and support throughout this process.
> > >
> > > Best regards,
> > >
> > > The Authors

---

> > > > ### Author Response · Authors · 2024-11-27
> > > > **Gratitude and Further Suggestions**
> > > >
> > > > Dear Reviewer 2k7C,
> > > >
> > > > We wish to express our deepest gratitude for your insightful comments and valuable suggestions, which have significantly enhanced the quality of our paper. As we are preparing to submit a new revision, we would be grateful if you could kindly provide any additional feedback or suggestions you may have. Addressing your concerns is of the utmost importance to us, and any further insights from you would be immensely beneficial in improving our work.
> > > >
> > > > Once again, we sincerely thank you for your thoughtful engagement and support throughout this process.
> > > >
> > > > Best regards,
> > > >
> > > > The Authors

---

> > ### Comment · Reviewer_2k7C · 2024-11-27
> >
> > I appreciate the additional experiments performed by the authors on more datasets and the added baseline for comparison, which addressed many of my concerns. Although I would not increase the score to 8, I think this is a good paper.

---

> > > ### Author Response · Authors · 2024-11-27
> > >
> > > Dear Reviewer 2k7C,
> > >
> > > Thank you very much for your valuable feedback. We are delighted to receive your approval of our work and to hear that you regard our paper as being of good quality.
> > >
> > > Thank you once again for your constructive insights, which have significantly enhanced our paper.
> > >
> > > Best regards,
> > >
> > > The Authors

---

### Official Review · Reviewer_VHTd · 2024-11-06

**Soundness:** 3
**Presentation:** 3
**Contribution:** 2
**Rating:** 8
**Confidence:** 3

**Summary:**

The manuscript proposes an improved electronic dynamical system---InstraTrain---to tackle the rapidly evolving nature of time series in a highly dynamic environment: it enables frequent model updates with microsecond-level intervals for real-world predictions.

The manuscript extends the extraordinary computational efficiency of the electronic dynamical system from inference to training, using two major components. It first formulates the time series prediction problem via the natural annealing procedure of a dynamical system, enabling performing efficient prediction. It further considers a physical realization of the proposed dynamical system.

**Strengths:**

* The manuscript is well-written. It starts from the statistical mechanics in physics and models the time-series prediction problem as a dynamical system. It then leverages the natural annealing process of a dynamical system to enable ultra-fast model training. Furthermore, it redesigns the electronic dynamical system from a hardware aspect, enabling the self-training feature. The reviewer enjoys reading the paper, especially section 3.
* The manuscript considers three types of baselines, covering 7 methods. It also uses an FEA software simulator to measure the accuracy and latency of the proposed method. Numerical results show that InstaTrain can achieve a significant speedup over all baseline models, while maintaining a $10^5$ reduction in energy consumption.

**Weaknesses:**

* The datasets used for the evaluation are not carefully explained, making the significance of the performance gain unclear. It would be great if the authors could elaborate on the dataset statistics. It would be great if the authors could include results on more challenging datasets.
* One intuition of the paper is that the algorithm could converge to the equilibrium state of the dynamical system. It would be great if the authors could justify this point, either theoretical or empirical.

**Questions:**

In lines 365-366, the authors claim that "the more accurate high-frequency update is unattainable due to the sluggish training speed of NP-GL, GNNs, and even SOTA online-training methods", as well as the "'Not achievable' are due to slow training" in the caption of Figure 5, the reviewer is a bit confused regarding these results, as they are not achievable. The authors are encouraged to explain this point.

---

> ### Author Response · Authors · 2024-11-21
> **Response to Reviewer VHTd (Part 1)**
>
> We sincerely appreciate your positive feedback and your constructive suggestions. In the following, we will address your questions one by one.
>
> **Add elaboration on dataset statistics and results on more challenging datasets.**
>
> Thanks for your valuable suggestion. In response, we have expanded the Appendix of the revised manuscript to include detailed statistics of the original datasets as well as two additional, more challenging datasets.  Given our focus on high-frequency data, we have incorporated two real-world datasets from a chemical detection platform, both collected at a sampling rate of 100 Hz:
>
> 1. Ammonia contains time series recordings from 72 metal-oxide sensors across six different locations under consistent wind speed and operating temperatures, resulting in a (12,755 × 432) array.
>
> 2. Toluene includes time series recordings from 72 sensors at one location under ten varying conditions (two wind speeds and five operating temperatures), forming a (12,854 × 720) array.
>
> To provide a more comprehensive comparison, we have also added three new Transformer-based baselines (Autoformer [1], DLinear [2], iTransformer [3]). Our evaluation now includes a total of eleven models across five datasets and three scenarios: static, low-frequency updates, and high-frequency updates. The evaluation section has been updated accordingly. The results are presented in the Table 1, Figure 5, and Figure 6 of the revised manuscript. We have attached the dataset statistics table and the accuracy comparison table below for your reference. Here, we present the performance of the best-performing GNN and Transformer-based model (Best Trans), a complete version of this table is provided in Table 5 in the Appendix of the revised manuscript.
>
> **Table:** Dataset Statistics
> | Dataset       | Carbon-Oxide | Methane | Stock  | Ammonia | Toluene |
> |---------------|--------------|---------|--------|---------|---------|
> | # of Samples  | 60000        | 60000   | 40060  | 12755   | 12854   |
> | # of Nodes    | 16           | 16      | 116    | 432     | 720     |
>
> **Table:** Accuracy comparison across datasets. LF / HF: Low / High Frequency. Online learning methods only have HF. In HF update, the results from the baselines are “Not Achievable” due to slow training.
> | Category      | Dataset      | Carbon-Oxide | Methane | Stock | Ammonia | Toluene |
> |---------------|--------------|--------------|---------|-------|---------|---------|
> | **Static**    | Best GNN     | 14.40        | 19.31   | 2.85  | 13.43   | 13.26   |
> |               | Best Trans   | 14.02        | 19.29   | 2.27  | 12.41   | 11.95   |
> |               | NPGL         | 13.90        | 19.22   | 2.01  | 12.15   | 11.43   |
> |               | InstaTrain   | 13.88        | 19.25   | 2.02  | 12.08   | 11.37   |
> | **LF Update** | Best GNN     | 10.28        | 11.57   | 1.70  | 4.72    | 5.19    |
> |               | Best Trans   | 8.53         | 9.72    | 1.22  | 3.94    | 4.85    |
> |               | NPGL         | 8.25         | 9.26    | 1.18  | 3.81    | 4.68    |
> |               | InstaTrain   | 8.28         | 9.22    | 1.20  | 3.72    | 4.67    |
> | **HF Update** | Best GNN     | *7.16*       | *7.36*  | *0.80* | *1.62*  | *2.11*  |
> |               | Best Trans   | *7.12*       | *7.25*  | *0.73* | *1.45*  | *1.94*  |
> |               | FSNet        | *7.11*       | *7.14*  | *0.79* | *1.48*  | *2.07*  |
> |               | PatchTST     | *7.05*       | *7.09*  | *0.80* | *1.46*  | *2.02*  |
> |               | OneNet       | *6.93*       | *7.11*  | *0.77* | *1.42*  | *1.93*  |
> |               | NPGL         | *6.81*       | *7.08*  | *0.68* | *1.39*  | *1.90*  |
> |               | InstaTrain   | 6.79         | 7.05    | 0.68  | 1.36    | 1.86    |
>
> *Note: Results marked in italics represent “Not Achievable” due to slow training.*
>
> [1] Wu, H., Xu, J., Wang, J., & Long, M. (2021). Autoformer: Decomposition transformers with auto-correlation for long-term series forecasting. Advances in neural information processing systems, 34, 22419-22430.
>
> [2] Zeng, A., Chen, M., Zhang, L., & Xu, Q. (2023). Are transformers effective for time series forecasting?. In Proceedings of the AAAI conference on artificial intelligence (Vol. 37, No. 9, pp. 11121-11128).
>
> [3] Liu, Y., Hu, T., Zhang, H., Wu, H., Wang, S., Ma, L., & Long, M. (2024). iTransformer: Inverted Transformers Are Effective for Time Series Forecasting. In The Twelfth International Conference on Learning Representations.

---

> > ### Author Response · Authors · 2024-11-21
> > **Response to Reviewer VHTd (Part 2)**
> >
> > **Justification on the convergence to the equilibrium state of the dynamical system**
> >
> > Thanks for your insightful question. We provide a detailed justification below, incorporating both theoretical analysis and empirical evidence from our experiments.
> >
> > The spontaneous energy decrease of the system is guaranteed by the designed node dynamics, which dictate how the system evolves over time. Specifically, the node dynamics are designed as $\frac{d\sigma_i}{dt} = -\frac{\partial H_{rv}}{\partial \sigma_i} = \sum_{j \neq i}^{N} \left( J_{ij} + J_{ji} \right) \sigma_j - 2 h_i \sigma_i.$
> > These dynamics are formulated based on Lyapunov stability principles, guaranteeing that the system evolves towards the lowest energy state $\frac{dH_{rv}}{d t} = \sum_{i=1}^{N} \left( \frac{\partial H_{rv}}{\partial \sigma_i} \frac{d \sigma_i}{dt} \right) \leq 0.$
> >
> > To empirically validate this, we have included a visualization of the system's energy with respect to annealing time in the revised manuscript's Appendix (Figure 7). The energy curve clearly demonstrates a convergence pattern, with the system’s energy rapidly decreasing and stabilizing over time as it approaches equilibrium. Furthermore, we introduce perturbations at the annealing time of 0.6e − 7s by adding Gaussian noise to the nodes at levels of 10% (blue curve), 20% (green curve), and 30% (red curve). These perturbations further confirm that the system can achieve equilibrium, as it returns to a stable state even when subjected to varying degrees of disturbance.
> >
> > **Explanation for the not achievable results**
> >
> > We are sorry for any confusions and appreciate the opportunity to clarity the meaning of “Not Achievable” results.
> >
> > To provide a comprehensive comparison, we evaluate InstaTrain alongside baseline models in three scenarios: static, low-frequency update, and high-frequency update. In the high-frequency update scenario, all models are set to update once every 100 snapshots, equivalent to a 1-second real-time interval. This setup requires models to complete parameter updates within 1 second to be ready for the next cycle. However, as shown in Figure 5, the training latency per snapshot for all baseline models exceeds 0.01 seconds, resulting in a cumulative training time of over 1 second for 100 snapshots. This delay means that baseline models cannot keep up with the high-frequency update schedule, making real-time adaptation “Not Achievable” for them in this scenario.
> > However, for the sake of comparison, we still calculate the accuracy of these baseline models under the high-frequency setup, assuming they could meet the update interval. This serves two purposes: (1) to emphasize the importance of high-frequency updates for improved performance on high-frequency datasets and (2) to provide a comprehensive comparison of accuracy across models.
> > We hope this clarifies the “Not Achievable” results and we have revised the manuscript to include this explanation for clarity.

---

### Author Response · Authors · 2024-12-03
**Summary of Rebuttal and Discussion**

Dear Area Chair and Reviewers,

We sincerely appreciate all the reviewers for their constructive feedback and insightful suggestions throughout the review process. We are pleased to note the positive outcomes from the discussion phase:

- Reviewer VHTd upheld the high score. Their suggestions are insightful, and we have addressed each point thoroughly with the responses incorporated in the revision.
- Reviewer 2k7C maintained their positive score, recognizing that our revisions effectively addressed their concerns. While the score was not raised to 8, their comments describing our paper as 'a good paper' are encouraging.
- Reviewer 3cwj increased the score following our clarifications on how InstaTrain addresses continuous data and the additional evaluation on other widely used datasets. We thank the reviewer for recognizing InstaTrain as a novel and promising approach.
- Reviewer hPWn provided valuable suggestions to add more baselines and additional evaluations on other widely used datasets. We have addressed each point thoroughly with our responses incorporated in the revision. We appreciate the reviewer for recognizing the high originality, quality, clarity, and significance of our work in their review.

As the discussion concludes, we would like to summarize the key points addressed:

1. Enhanced Comparison and Evaluation.
- In response to Reviewer VHTd and Reviewer 2k7C, we conducted further experiments on more challenging datasets. The results, detailed in Table 1, Figure 5, and Figure 6, highlight InstaTrain's superior performance on high-dimensional datasets.
- Addressing Reviewer hPWn’s request, we included three new SOTA baselines in Table 1, Figure 5, and Figure 6, further validating the strengths of InstaTrain.
- Following the suggestions of Reviewer 3cwj and Reviewer hPWn, we evaluated InstaTrain’s performance on additional widely used datasets, as presented in Appendix A.3, providing a more comprehensive evaluation.
2. Improved Explanation and Presentation.
- To address the question from Reviewer VHTd and Reviewer 2k7C regarding the "Not Achievable" results, we provided a detailed explanation of why baseline methods fail to achieve optimal results in high-frequency scenarios, thereby highlighting InstaTrain’s advantages in these settings.
- Responding to Reviewer VHTd’s request, we included both theoretical and empirical justifications for the convergence to the equilibrium state within the dynamical system.
- In response to Reviewer 2k7C’s inquiry, we provided a detailed explanation of the calculations involved in energy reduction, ensuring clarity.
- To address Reviewer 3cwj’s questions, we elaborated on how InstaTrain effectively manages continuous real-valued data within the Background section, ensuring a comprehensive understanding of our approach.
- Following Reviewer hPWn’s recommendation, we added more tables to present our results, facilitating easier comparison and interpretation of the data.

We would like to emphasize the **contributions** of InstaTrain as recognized by reviewers:

- **A new fast and low-cost training method:** InstaTrain enables a novel type of efficient training approach by transforming the traditional expensive and slow training process on digital processors into fast and low-cost natural annealing processes within dynamical systems, achieving orders-of-magnitude speedup in model training and updating compared to baselines.

- **Expanding the capability of an emerging AI processor:** InstaTrain extends the exceptional computational potential of the emerging dynamical system computing substrate from inference-only to training, opening a new research direction for efficient AI training.

- **Improving accuracy for high-frequency ML challenges:** With the microsecond-level model update capability, InstaTrain provides significantly improved accuracy, particularly in tackling complex high-frequency ML problems.

Finally, we thank the reviewers for their invaluable comments, which have strengthened our work. We hope InstaTrain provides meaningful insights and advances the development of high-speed and low-cost AI training.

Best Regards,

The Authors

---

### Meta-Review · Area_Chair_dATn · 2024-12-19

**Metareview:**

This paper outlines a system for time-series model training in environments where data distributions can change within microseconds. The authors further augment existing electronic hardware with parameter update modules to enable both training and inference on the same system. The experimental results reported on several high-frequency datasets including stock market and chemical sensor data, show improvements over the state-of-the-art. Most reviewers voted for acceptance. The authors provided an extensive rebuttal to address the concerns raised by the reviewers. I would encourage the authors to incorporate those results and insights in the final version/additional appendices to further strengthen the paper.

**Additional Comments On Reviewer Discussion:**

Reviewers acknowledged the response provided by the authors. Several adjusted their scores based the detailed clarifications, including producing extensive additional empirical evidence. Post rebuttal, only one reviewer out of the 4 voted for weak reject, while all others increased their scores reflecting their positive assessment of the paper and indicating this work was worth sharing with the community.

---

### Decision · Program_Chairs · 2025-01-22

Accept (Poster)